# SAGA: A Fast Incremental Gradient Method With Support for Non-Strongly Convex Composite Objectives

**Aaron Defazio**
Ambiata *
Australian National University, Canberra

**Francis Bach**
INRIA - Sierra Project-Team
École Normale Supérieure, Paris, France

**Simon Lacoste-Julien**
INRIA - Sierra Project-Team
École Normale Supérieure, Paris, France

## Abstract

In this work we introduce a new optimisation method called SAGA in the spirit of SAG, SDCA, MISO and SVRG, a set of recently proposed incremental gradient algorithms with fast linear convergence rates. SAGA improves on the theory behind SAG and SVRG, with better theoretical convergence rates, and has support for composite objectives where a proximal operator is used on the regulariser. Unlike SDCA, SAGA supports non-strongly convex problems directly, and is adaptive to any inherent strong convexity of the problem. We give experimental results showing the effectiveness of our method.

## 1 Introduction

Remarkably, recent advances [1, 2] have shown that it is possible to minimise strongly convex finite sums provably faster in expectation than is possible without the finite sum structure. This is significant for machine learning problems as a finite sum structure is common in the empirical risk minimisation setting. The requirement of strong convexity is likewise satisfied in machine learning problems in the typical case where a quadratic regulariser is used.

In particular, we are interested in minimising functions of the form

$$f(x) = \frac{1}{n} \sum_{i=1}^{n} f_i(x),$$

where $x \in \mathbb{R}^d$, each $f_i$ is convex and has Lipschitz continuous derivatives with constant $L$. We will also consider the case where each $f_i$ is strongly convex with constant $\mu$, and the "composite" (or proximal) case where an additional regularisation function is added:

$$F(x) = f(x) + h(x),$$

where $h \colon \mathbb{R}^d \to \mathbb{R}^d$ is convex but potentially non-differentiable, and where the proximal operation of $h$ is easy to compute — few incremental gradient methods are applicable in this setting [3][4].

Our contributions are as follows. In Section 2 we describe the SAGA algorithm, a novel incremental gradient method. In Section 5 we prove theoretical convergence rates for SAGA in the strongly convex case better than those for SAG [1] and SVRG [5], and a factor of 2 from the SDCA [2] convergence rates. These rates also hold in the composite setting. Additionally, we show that

like SAG but unlike SDCA, our method is applicable to non-strongly convex problems without modification. We establish theoretical convergence rates for this case also. In Section 3 we discuss the relation between each of the fast incremental gradient methods, showing that each stems from a very small modification of another.

## 2 SAGA Algorithm

We start with some known initial vector $x^0 \in \mathbb{R}^d$ and known derivatives $f_i'(\phi_i^0) \in \mathbb{R}^d$ with $\phi_i^0 = x^0$ for each $i$. These derivatives are stored in a table data-structure of length $n$, or alternatively a $n \times d$ matrix. For many problems of interest, such as binary classification and least-squares, only a single floating point value instead of a full gradient vector needs to be stored (see Section 4). SAGA is inspired both from SAG [1] and SVRG [5] (as we will discuss in Section 3). SAGA uses a step size of $\gamma$ and makes the following updates, starting with $k = 0$:

---

**SAGA Algorithm:** Given the value of $x^k$ and of each $f_i'(\phi_i^k)$ at the end of iteration $k$, the updates for iteration $k + 1$ is as follows:

1. Pick a $j$ uniformly at random.

2. Take $\phi_j^{k+1} = x^k$, and store $f_j'(\phi_j^{k+1})$ in the table. All other entries in the table remain unchanged. The quantity $\phi_j^{k+1}$ is not explicitly stored.

3. Update $x$ using $f_j'(\phi_j^{k+1})$, $f_j'(\phi_j^k)$ and the table average:

$$w^{k+1} = x^k - \gamma \left[ f_j'(\phi_j^{k+1}) - f_j'(\phi_j^k) + \frac{1}{n} \sum_{i=1}^n f_i'(\phi_i^k) \right], \tag{1}$$

$$x^{k+1} = \mathrm{prox}_\gamma^h \left( w^{k+1} \right). \tag{2}$$

---

The proximal operator we use above is defined as

$$\mathrm{prox}_\gamma^h (y) := \operatorname*{argmin}_{x \in \mathbb{R}^d} \left\{ h(x) + \frac{1}{2\gamma} \|x - y\|^2 \right\}. \tag{3}$$

In the strongly convex case, when a step size of $\gamma = 1/(2(\mu n + L))$ is chosen, we have the following convergence rate in the composite and hence also the non-composite case:

$$\mathbb{E} \left\| x^k - x^* \right\|^2 \le \left( 1 - \frac{\mu}{2(\mu n + L)} \right)^k \left[ \left\| x^0 - x^* \right\|^2 + \frac{n}{\mu n + L} \left[ f(x^0) - \langle f'(x^*), x^0 - x^* \rangle - f(x^*) \right] \right].$$

We prove this result in Section 5. The requirement of strong convexity can be relaxed from needing to hold for each $f_i$ to just holding on average, but at the expense of a worse geometric rate $(1 - \frac{\mu}{6(\mu n + L)})$, requiring a step size of $\gamma = 1/(3(\mu n + L))$.

In the non-strongly convex case, we have established the convergence rate in terms of the average iterate, excluding step 0: $\bar{x}^k = \frac{1}{k} \sum_{t=1}^k x^t$. Using a step size of $\gamma = 1/(3L)$ we have

$$\mathbb{E} \left[ F(\bar{x}^k) \right] - F(x^*) \le \frac{4n}{k} \left[ \frac{2L}{n} \left\| x^0 - x^* \right\|^2 + f(x^0) - \langle f'(x^*), x^0 - x^* \rangle - f(x^*) \right].$$

This result is proved in the supplementary material. Importantly, when this step size $\gamma = 1/(3L)$ is used, our algorithm *automatically adapts* to the level of strong convexity $\mu > 0$ naturally present, giving a convergence rate of (see the comment at the end of the proof of Theorem 1):

$$\mathbb{E} \left\| x^k - x^* \right\|^2 \le \left( 1 - \min \left\{ \frac{1}{4n}, \frac{\mu}{3L} \right\} \right)^k \left[ \left\| x^0 - x^* \right\|^2 + \frac{2n}{3L} \left[ f(x^0) - \langle f'(x^*), x^0 - x^* \rangle - f(x^*) \right] \right].$$

Although any incremental gradient method can be applied to non-strongly convex problems via the addition of a small quadratic regularisation, the amount of regularisation is an additional tunable parameter which our method avoids.

## 3 Related Work

We explore the relationship between SAGA and the other fast incremental gradient methods in this section. By using SAGA as a midpoint, we are able to provide a more unified view than is available in the existing literature. A brief summary of the properties of each method considered in this section is given in Figure 1. The method from [3], which handles the non-composite setting, is not listed as its rate is of the slow type and can be up to $n$ times smaller than the one for SAGA or SVRG [5].

|  | SAGA | SAG | SDCA | SVRG | FINITO |
|---|:---:|:---:|:---:|:---:|:---:|
| Strongly Convex (SC) | ✓ | ✓ | ✓ | ✓ | ✓ |
| Convex, Non-SC* | ✓ | ✓ | ✗ | ? | ? |
| Prox Reg. | ✓ | ? | ✓[6] | ✓ | ✗ |
| Non-smooth | ✗ | ✗ | ✓ | ✗ | ✗ |
| Low Storage Cost | ✗ | ✗ | ✗ | ✓ | ✗ |
| Simple(-ish) Proof | ✓ | ✗ | ✓ | ✓ | ✓ |
| Adaptive to SC | ✓ | ✓ | ✗ | ? | ? |

Figure 1: Basic summary of method properties. Question marks denote unproven, but not experimentally ruled out cases. (*) Note that any method can be applied to non-strongly convex problems by adding a small amount of L2 regularisation, this row describes methods that do not require this trick.

**SAGA: midpoint between SAG and SVRG/S2GD**

In [5], the authors make the observation that the variance of the standard stochastic gradient (SGD) update direction can only go to zero if decreasing step sizes are used, thus preventing a linear convergence rate unlike for batch gradient descent. They thus propose to use a variance reduction approach (see [7] and references therein for example) on the SGD update in order to be able to use constant step sizes and get a linear convergence rate. We present the updates of their method called SVRG (Stochastic Variance Reduced Gradient) in (6) below, comparing it with the non-composite form of SAGA rewritten in (5). They also mention that SAG (Stochastic Average Gradient) [1] can be interpreted as reducing the variance, though they do not provide the specifics. Here, we make this connection clearer and relate it to SAGA.

We first review a slightly more generalized version of the variance reduction approach (we allow the updates to be biased). Suppose that we want to use Monte Carlo samples to estimate $\mathbb{E}X$ and that we can compute efficiently $\mathbb{E}Y$ for another random variable $Y$ that is highly correlated with $X$. One variance reduction approach is to use the following estimator $\theta_\alpha$ as an approximation to $\mathbb{E}X$: $\theta_\alpha := \alpha(X-Y)+\mathbb{E}Y$, for a step size $\alpha \in [0,1]$. We have that $\mathbb{E}\theta_\alpha$ is a convex combination of $\mathbb{E}X$ and $\mathbb{E}Y$: $\mathbb{E}\theta_\alpha = \alpha\mathbb{E}X + (1-\alpha)\mathbb{E}Y$. The standard variance reduction approach uses $\alpha = 1$ and the estimate is unbiased $\mathbb{E}\theta_1 = \mathbb{E}X$. The variance of $\theta_\alpha$ is: $\text{Var}(\theta_\alpha) = \alpha^2[\text{Var}(X) + \text{Var}(Y) - 2\,\text{Cov}(X,Y)]$, and so if $\text{Cov}(X,Y)$ is big enough, the variance of $\theta_\alpha$ is reduced compared to $X$, giving the method its name. By varying $\alpha$ from 0 to 1, we increase the variance of $\theta_\alpha$ towards its maximum value (which usually is still smaller than the one for $X$) while decreasing its bias towards zero.

Both SAGA and SAG can be derived from such a variance reduction viewpoint: here $X$ is the SGD direction sample $f'_j(x^k)$, whereas $Y$ is a past stored gradient $f'_j(\phi_j^k)$. SAG is obtained by using $\alpha = 1/n$ (update rewritten in our notation in (4)), whereas SAGA is the unbiased version with $\alpha = 1$ (see (5) below). For the same $\phi$'s, the variance of the SAG update is $1/n^2$ times the one of SAGA, but at the expense of having a non-zero bias. This non-zero bias might explain the complexity of the convergence proof of SAG and why the theory has not yet been extended to proximal operators. By using an unbiased update in SAGA, we are able to obtain a simple and tight theory, with better constants than SAG, as well as theoretical rates for the use of proximal operators.

$$\text{(SAG)} \qquad x^{k+1} = x^k - \gamma\left[\frac{f'_j(x^k) - f'_j(\phi_j^k)}{n} + \frac{1}{n}\sum_{i=1}^n f'_i(\phi_i^k)\right], \qquad (4)$$

$$\text{(SAGA)} \qquad x^{k+1} = x^k - \gamma\left[f'_j(x^k) - f'_j(\phi_j^k) + \frac{1}{n}\sum_{i=1}^n f'_i(\phi_i^k)\right], \qquad (5)$$

$$\text{(SVRG)} \qquad x^{k+1} = x^k - \gamma\left[f'_j(x^k) - f'_j(\tilde{x}) + \frac{1}{n}\sum_{i=1}^n f'_i(\tilde{x})\right]. \qquad (6)$$

The SVRG update (6) is obtained by using $Y = f'_j(\tilde{x})$ with $\alpha = 1$ (and is thus unbiased – we note that SAG is the only method that we present in the related work that has a biased update direction). The vector $\tilde{x}$ is not updated every step, but rather the loop over $k$ appears inside an outer loop, where $\tilde{x}$ is updated at the start of each outer iteration. Essentially SAGA is at the midpoint between SVRG and SAG; it updates the $\phi_j$ value each time index $j$ is picked, whereas SVRG updates all of $\phi$'s as a batch. The S2GD method [8] has the same update as SVRG, just differing in how the number of inner loop iterations is chosen. We use SVRG henceforth to refer to both methods.

SVRG makes a trade-off between time and space. For the equivalent practical convergence rate it makes 2x-3x more gradient evaluations, but in doing so it does not need to store a table of gradients, but a single average gradient. The usage of SAG vs. SVRG is problem dependent. For example for linear predictors where gradients can be stored as a reduced vector of dimension $p - 1$ for $p$ classes, SAGA is preferred over SVRG both theoretically and in practice. For neural networks, where no theory is available for either method, the storage of gradients is generally more expensive than the additional backpropagations, but this is computer architecture dependent.

SVRG also has an additional parameter besides step size that needs to be set, namely the number of iterations per inner loop ($m$). This parameter can be set via the theory, or conservatively as $m = n$, however doing so does not give anywhere near the best practical performance. Having to tune one parameter instead of two is a practical advantage for SAGA.

**Finito/MISO$\mu$**

To make the relationship with other prior methods more apparent, we can rewrite the SAGA algorithm (in the non-composite case) in term of an additional intermediate quantity $u^k$, with $u^0 := x^0 + \gamma \sum_{i=1}^{n} f_i'(x^0)$, in addition to the usual $x^k$ iterate as described previously:

---

**SAGA: Equivalent reformulation for non-composite case:** Given the value of $u^k$ and of each $f_i'(\phi_i^k)$ at the end of iteration $k$, the updates for iteration $k + 1$, is as follows:

1. Calculate $x^k$:
$$x^k = u^k - \gamma \sum_{i=1}^{n} f_i'(\phi_i^k). \tag{7}$$

2. Update $u$ with $u^{k+1} = u^k + \frac{1}{n}(x^k - u^k)$.

3. Pick a $j$ uniformly at random.

4. Take $\phi_j^{k+1} = x^k$, and store $f_j'(\phi_j^{k+1})$ in the table replacing $f_j'(\phi_j^k)$. All other entries in the table remain unchanged. The quantity $\phi_j^{k+1}$ is not explicitly stored.

---

Eliminating $u^k$ recovers the update (5) for $x^k$. We now describe how the Finito [9] and MISO$\mu$ [10] methods are closely related to SAGA. Both Finito and MISO$\mu$ use updates of the following form, for a step length $\gamma$:
$$x^{k+1} = \frac{1}{n} \sum_i \phi_i^k - \gamma \sum_{i=1}^{n} f_i'(\phi_i^k). \tag{8}$$

The step size used is of the order of $1/\mu n$. To simplify the discussion of this algorithm we will introduce the notation $\bar{\phi} = \frac{1}{n} \sum_i \phi_i^k$.

SAGA can be interpreted as Finito, but with the quantity $\bar{\phi}$ replaced with $u$, which is updated in the same way as $\bar{\phi}$, but *in expectation*. To see this, consider how $\bar{\phi}$ changes in expectation:
$$\mathbb{E}\left[\bar{\phi}^{k+1}\right] = \mathbb{E}\left[\bar{\phi}^k + \frac{1}{n}\left(x^k - \phi_j^k\right)\right] = \bar{\phi}^k + \frac{1}{n}\left(x^k - \bar{\phi}^k\right).$$

The update is identical in expectation to the update for $u$, $u^{k+1} = u^k + \frac{1}{n}(x^k - u^k)$. There are three advantages of SAGA over Finito/MISO$\mu$. SAGA does not require strong convexity to work, it has support for proximal operators, and it does not require storing the $\phi_i$ values. MISO has proven support for proximal operators only in the case where impractically small step sizes are used [10]. The big advantage of Finito/MISO$\mu$ is that when using a per-pass re-permuted access ordering, empirical speed-ups of up-to a factor of 2x has been observed. This access order can also be used with the other methods discussed, but with smaller empirical speed-ups. Finito/MISO$\mu$ is particularly useful when $f_i$ is computationally expensive to compute compared to the extra storage costs required over the other methods.

**SDCA**

The Stochastic Dual Coordinate Descent (SDCA) [2] method on the surface appears quite different from the other methods considered. It works with the convex conjugates of the $f_i$ functions. However, in this section we show a novel transformation of SDCA into an equivalent method that only works with primal quantities, and is closely related to the MISO$\mu$ method.

Consider the following algorithm:

---

**SDCA algorithm in the primal**

Step $k + 1$:

1. Pick an index $j$ uniformly at random.
2. Compute $\phi_j^{k+1} = \operatorname{prox}_\gamma^{f_j}(z)$, where $\gamma = \frac{1}{\mu n}$ and $z = -\gamma \sum_{i \neq j}^n f_i'(\phi_i^k)$.
3. Store the gradient $f_j'(\phi_j^{k+1}) = \frac{1}{\gamma}\left(z - \phi_j^{k+1}\right)$ in the table at location $j$. For $i \neq j$, the table entries are unchanged ($f_i'(\phi_i^{k+1}) = f_i'(\phi_i^k)$).

At completion, return $x^k = -\gamma \sum_i^n f_i'(\phi_i^k)$.

---

We claim that this algorithm is equivalent to the version of SDCA where exact block-coordinate maximisation is used on the dual.[1] Firstly, note that while SDCA was originally described for one-dimensional outputs (binary classification or regression), it has been expanded to cover the multi-class predictor case [11] (called Prox-SDCA there). In this case, the primal objective has a separate strongly convex regulariser, and the functions $f_i$ are restricted to the form $f_i(x) := \psi_i(X_i^T x)$, where $X_i$ is a $d \times p$ feature matrix, and $\psi_i$ is the loss function that takes a $p$ dimensional input, for $p$ classes. To stay in the same general setting as the other incremental gradient methods, we work directly with the $f_i(x)$ functions rather than the more structured $\psi_i(X_i^T x)$. The dual objective to maximise then becomes

$$D(\alpha) = \left[ -\frac{\mu}{2} \left\| \frac{1}{\mu n} \sum_{i=1}^n \alpha_i \right\|^2 - \frac{1}{n} \sum_{i=1}^n f_i^*(-\alpha_i) \right],$$

where $\alpha_i$'s are $d$-dimensional dual variables. Generalising the exact block-coordinate maximisation update that SDCA performs to this form, we get the dual update for block $j$ (with $x^k$ the current primal iterate):

$$\alpha_j^{k+1} = \alpha_j^k + \operatorname*{argmax}_{\Delta a_j \in \mathbb{R}^d} \left\{ -f_j^*\left(-\alpha_j^k - \Delta\alpha_j\right) - \frac{\mu n}{2} \left\| x^k + \frac{1}{\mu n}\Delta\alpha_j \right\|^2 \right\}. \tag{9}$$

In the special case where $f_i(x) = \psi_i(X_i^T x)$, we can see that (9) gives exactly the same update as Option I of Prox-SDCA in [11, Figure 1], which operates instead on the equivalent $p$-dimensional dual variables $\tilde{\alpha}_i$ with the relationship that $\alpha_i = X_i\tilde{\alpha}_i$.[2] As noted by Shalev-Shwartz & Zhang [11], the update (9) is actually an instance of the proximal operator of the convex conjugate of $f_j$. Our primal formulation exploits this fact by using a relation between the proximal operator of a function and its convex conjugate known as the Moreau decomposition:

$$\operatorname{prox}^{f^*}(v) = v - \operatorname{prox}^f(v).$$

This decomposition allows us to compute the proximal operator of the conjugate via the primal proximal operator. As this is the only use in the basic SDCA method of the conjugate function, applying this decomposition allows us to completely eliminate the "dual" aspect of the algorithm, yielding the above primal form of SDCA. The dual variables are related to the primal representatives $\phi_i$'s through $\alpha_i = -f_i'(\phi_i)$. The KKT conditions ensure that if the $\alpha_i$ values are dual optimal then $x^k = \gamma \sum_i \alpha_i$ as defined above is primal optimal. The same trick is commonly used to interpret Dijkstra's set intersection as a primal algorithm instead of a dual block coordinate descent algorithm [12].

The primal form of SDCA differs from the other incremental gradient methods described in this section in that it assumes strong convexity is induced by a separate strongly convex regulariser, rather than each $f_i$ being strongly convex. In fact, SDCA can be modified to work without a separate regulariser, giving a method that is at the midpoint between Finito and SDCA. We detail such a method in the supplementary material.

**SDCA variants**

The SDCA theory has been expanded to cover a number of other methods of performing the coordinate step [11]. These variants replace the proximal operation in our primal interpretation in the previous section with an update where $\phi_j^{k+1}$ is chosen so that: $f_j'(\phi_j^{k+1}) = (1-\beta)f_j'(\phi_j^k) + \beta f_j'(x^k)$, where $x^k = -\frac{1}{\mu n}\sum_i f_i'(\phi_i^k)$. The variants differ in how $\beta \in [0,1]$ is chosen. Note that $\phi_j^{k+1}$ does not actually have to be explicitly known, just the gradient $f_j'(\phi_j^{k+1})$, which is the result of the above interpolation. Variant 5 by Shalev-Shwartz & Zhang [11] does not require operations on the conjugate function, it simply uses $\beta = \frac{\mu n}{L + \mu n}$. The most practical variant performs a line search involving the convex conjugate to determine $\beta$. As far as we are aware, there is no simple primal equivalent of this line search. So in cases where we can not compute the proximal operator from the standard SDCA variant, we can either introduce a tuneable parameter into the algorithm ($\beta$), or use a dual line search, which requires an efficient way to evaluate the convex conjugates of each $f_i$.

## 4 Implementation

We briefly discuss some implementation concerns:

- For many problems each derivative $f_i'$ is just a simple weighting of the $i$th data vector. Logistic regression and least squares have this property. In such cases, instead of storing the full derivative $f_i'$ for each $i$, we need only to store the weighting constants. This reduces the storage requirements to be the same as the SDCA method in practice. A similar trick can be applied to multi-class classifiers with $p$ classes by storing $p-1$ values for each $i$.

- Our algorithm assumes that initial gradients are known for each $f_i$ at the starting point $x^0$. Instead, a heuristic may be used where during the first pass, data-points are introduced one-by-one, in a non-randomized order, with averages computed in terms of those data-points processed so far. This procedure has been successfully used with SAG [1].

- The SAGA update as stated is slower than necessary when derivatives are sparse. A just-in-time updating of $u$ or $x$ may be performed just as is suggested for SAG [1], which ensures that only sparse updates are done at each iteration.

- We give the form of SAGA for the case where each $f_i$ is strongly convex. However in practice we usually have only convex $f_i$, with strong convexity in $f$ induced by the addition of a quadratic regulariser. This quadratic regulariser may be split amongst the $f_i$ functions evenly, to satisfy our assumptions. It is perhaps easier to use a variant of SAGA where the regulariser $\frac{\mu}{2}||x||^2$ is explicit, such as the following modification of Equation (5):

$$x^{k+1} = (1-\gamma\mu)\,x^k - \gamma\left[f_j'(x^k) - f_j'(\phi_j^k) + \frac{1}{n}\sum_i f_i'(\phi_i^k)\right].$$

  For sparse implementations instead of scaling $x^k$ at each step, a separate scaling constant $\beta^k$ may be scaled instead, with $\beta^k x^k$ being used in place of $x^k$. This is a standard trick used with stochastic gradient methods.

For sparse problems with a quadratic regulariser the just-in-time updating can be a little intricate. In the supplementary material we provide example python code showing a correct implementation that uses each of the above tricks.

## 5 Theory

In this section, all expectations are taken with respect to the choice of $j$ at iteration $k+1$ and conditioned on $x^k$ and each $f_i'(\phi_i^k)$ unless stated otherwise.

We start with two basic lemmas that just state properties of convex functions, followed by Lemma 1, which is specific to our algorithm. The proofs of each of these lemmas is in the supplementary material.

**Lemma 1.** *Let $f(x) = \frac{1}{n}\sum_{i=1}^n f_i(x)$. Suppose each $f_i$ is $\mu$-strongly convex and has Lipschitz continuous gradients with constant L. Then for all $x$ and $x^*$:*

$$\langle f'(x), x^* - x\rangle \le \frac{L-\mu}{L}\left[f(x^*) - f(x)\right] - \frac{\mu}{2}\left\|x^* - x\right\|^2$$

$$-\frac{1}{2Ln}\sum_i \|f_i'(x^*) - f_i'(x)\|^2 - \frac{\mu}{L}\left\langle f'(x^*), x - x^* \right\rangle.$$

**Lemma 2.** *We have that for all $\phi_i$ and $x^*$:*

$$\frac{1}{n}\sum_i \|f_i'(\phi_i) - f_i'(x^*)\|^2 \leq 2L\left[\frac{1}{n}\sum_i f_i(\phi_i) - f(x^*) - \frac{1}{n}\sum_i \left\langle f_i'(x^*), \phi_i - x^* \right\rangle\right].$$

**Lemma 3.** *It holds that for any $\phi_i^k$, $x^*$, $x^k$ and $\beta > 0$, with $w^{k+1}$ as defined in Equation 1:*

$$\mathbb{E}\left\|w^{k+1} - x^k - \gamma f'(x^*)\right\|^2 \leq \gamma^2(1+\beta^{-1})\mathbb{E}\left\|f_j'(\phi_j^k) - f_j'(x^*)\right\|^2 + \gamma^2(1+\beta)\mathbb{E}\left\|f_j'(x^k) - f_j'(x^*)\right\|^2$$
$$- \gamma^2\beta\left\|f'(x^k) - f'(x^*)\right\|^2.$$

**Theorem 1.** *With $x^*$ the optimal solution, define the Lyapunov function $T$ as:*

$$T^k := T(x^k, \{\phi_i^k\}_{i=1}^n) := \frac{1}{n}\sum_i f_i(\phi_i^k) - f(x^*) - \frac{1}{n}\sum_i \left\langle f_i'(x^*), \phi_i^k - x^* \right\rangle + c\left\|x^k - x^*\right\|^2.$$

*Then with $\gamma = \frac{1}{2(\mu n + L)}$, $c = \frac{1}{2\gamma(1-\gamma\mu)n}$, and $\kappa = \frac{1}{\gamma\mu}$, we have the following expected change in the Lyapunov function between steps of the SAGA algorithm (conditional on $T^k$):*

$$\mathbb{E}[T^{k+1}] \leq (1 - \frac{1}{\kappa})T^k.$$

*Proof.* The first three terms in $T^{k+1}$ are straight-forward to simplify:

$$\mathbb{E}\left[\frac{1}{n}\sum_i f_i(\phi_i^{k+1})\right] = \frac{1}{n}f(x^k) + \left(1 - \frac{1}{n}\right)\frac{1}{n}\sum_i f_i(\phi_i^k).$$

$$\mathbb{E}\left[-\frac{1}{n}\sum_i \left\langle f_i'(x^*), \phi_i^{k+1} - x^* \right\rangle\right] = -\frac{1}{n}\left\langle f'(x^*), x^k - x^* \right\rangle - \left(1 - \frac{1}{n}\right)\frac{1}{n}\sum_i \left\langle f_i'(x^*), \phi_i^k - x^* \right\rangle.$$

For the change in the last term of $T^{k+1}$, we apply the non-expansiveness of the proximal operator[3]:

$$c\left\|x^{k+1} - x^*\right\|^2 = c\left\|\mathrm{prox}_\gamma(w^{k+1}) - \mathrm{prox}_\gamma(x^* - \gamma f'(x^*))\right\|^2$$
$$\leq c\left\|w^{k+1} - x^* + \gamma f'(x^*)\right\|^2.$$

We expand the quadratic and apply $\mathbb{E}[w^{k+1}] = x^k - \gamma f'(x^k)$ to simplify the inner product term:

$$c\mathbb{E}\left\|w^{k+1} - x^* + \gamma f'(x^*)\right\|^2 = c\mathbb{E}\left\|x^k - x^* + w^{k+1} - x^k + \gamma f'(x^*)\right\|^2$$
$$= c\left\|x^k - x^*\right\|^2 + 2c\mathbb{E}\left[\left\langle w^{k+1} - x^k + \gamma f'(x^*), x^k - x^* \right\rangle\right] + c\mathbb{E}\left\|w^{k+1} - x^k + \gamma f'(x^*)\right\|^2$$
$$= c\left\|x^k - x^*\right\|^2 - 2c\gamma\left\langle f'(x^k) - f'(x^*), x^k - x^* \right\rangle + c\mathbb{E}\left\|w^{k+1} - x^k + \gamma f'(x^*)\right\|^2$$
$$\leq c\left\|x^k - x^*\right\|^2 - 2c\gamma\left\langle f'(x^k), x^k - x^* \right\rangle + 2c\gamma\left\langle f'(x^*), x^k - x^* \right\rangle - c\gamma^2\beta\left\|f'(x^k) - f'(x^*)\right\|^2$$
$$+ \left(1+\beta^{-1}\right)c\gamma^2\mathbb{E}\left\|f_j'(\phi_j^k) - f_j'(x^*)\right\|^2 + (1+\beta)c\gamma^2\mathbb{E}\left\|f_j'(x^k) - f_j'(x^*)\right\|^2. \quad \text{(Lemma 3)}$$

The value of $\beta$ shall be fixed later. Now we apply Lemma 1 to bound $-2c\gamma\left\langle f'(x^k), x^k - x^* \right\rangle$ and Lemma 2 to bound $\mathbb{E}\left\|f_j'(\phi_j^k) - f_j'(x^*)\right\|^2$:

$$c\mathbb{E}\left\|x^{k+1} - x^*\right\|^2 \leq (c - c\gamma\mu)\left\|x^k - x^*\right\|^2 + \left((1+\beta)c\gamma^2 - \frac{c\gamma}{L}\right)\mathbb{E}\left\|f_j'(x^k) - f_j'(x^*)\right\|^2$$
$$- \frac{2c\gamma(L-\mu)}{L}\left[f(x^k) - f(x^*) - \left\langle f'(x^*), x^k - x^* \right\rangle\right] - c\gamma^2\beta\left\|f'(x^k) - f'(x^*)\right\|^2$$
$$+ 2\left(1+\beta^{-1}\right)c\gamma^2 L\left[\frac{1}{n}\sum_i f_i(\phi_i^k) - f(x^*) - \frac{1}{n}\sum_i \left\langle f_i'(x^*), \phi_i^k - x^* \right\rangle\right].$$

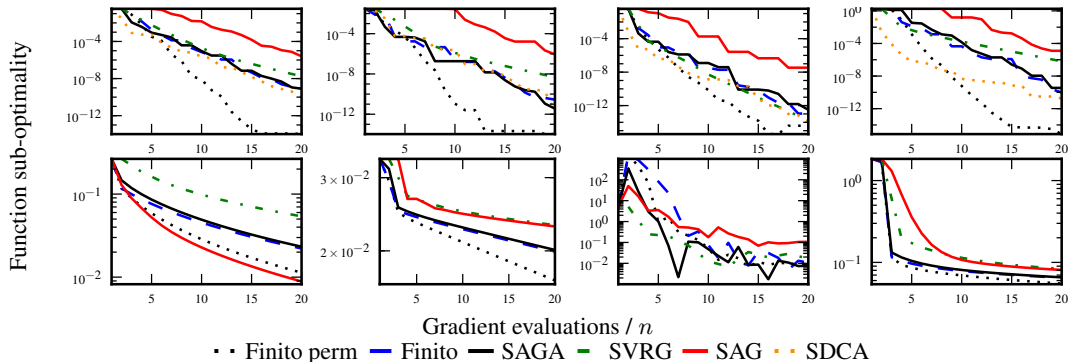

**Figure 2:** From left to right we have the MNIST, COVTYPE, IJCNN1 and MILLIONSONG datasets. Top row is the L2 regularised case, bottom row the L1 regularised case.

We can now combine the bounds that we have derived for each term in $T$, and pull out a fraction $\frac{1}{\kappa}$ of $T^k$ (for any $\kappa$ at this point). Together with the inequality $-\left\|f'(x^k) - f'(x^*)\right\|^2 \leq -2\mu\left[f(x^k) - f(x^*) - \left\langle f'(x^*), x^k - x^*\right\rangle\right]$ [13, Thm. 2.1.10], that yields:

$$\mathbb{E}[T^{k+1}] - T^k \leq -\frac{1}{\kappa}T^k + \left(\frac{1}{n} - \frac{2c\gamma(L-\mu)}{L} - 2c\gamma^2\mu\beta\right)\left[f(x^k) - f(x^*) - \left\langle f'(x^*), x^k - x^*\right\rangle\right]$$

$$+ \left(\frac{1}{\kappa} + 2(1+\beta^{-1})c\gamma^2 L - \frac{1}{n}\right)\left[\frac{1}{n}\sum_i f_i(\phi_i^k) - f(x^*) - \frac{1}{n}\sum_i\left\langle f_i'(x^*), \phi_i^k - x^*\right\rangle\right]$$

$$+ \left(\frac{1}{\kappa} - \gamma\mu\right)c\left\|x^k - x^*\right\|^2 + \left((1+\beta)\gamma - \frac{1}{L}\right)c\gamma\mathbb{E}\left\|f_j'(x^k) - f_j'(x^*)\right\|^2. \tag{10}$$

Note that each of the terms in square brackets are positive, and it can be readily verified that our assumed values for the constants ($\gamma = \frac{1}{2(\mu n + L)}$, $c = \frac{1}{2\gamma(1-\gamma\mu)n}$, and $\kappa = \frac{1}{\gamma\mu}$), together with $\beta = \frac{2\mu n + L}{L}$ ensure that each of the quantities in round brackets are non-positive (the constants were determined by setting all the round brackets to zero except the second one — see [14] for the details). **Adaptivity to strong convexity result:** Note that when using the $\gamma = \frac{1}{3L}$ step size, the same $c$ as above can be used with $\beta = 2$ and $\frac{1}{\kappa} = \min\left\{\frac{1}{4n}, \frac{\mu}{3L}\right\}$ to ensure non-positive terms. $\qquad\square$

**Corollary 1.** *Note that $c\left\|x^k - x^*\right\|^2 \leq T^k$, and therefore by chaining the expectations, plugging in the constants explicitly and using $\mu(n-0.5) \leq \mu n$ to simplify the expression, we get:*

$$\mathbb{E}\left[\left\|x^k - x^*\right\|^2\right] \leq \left(1 - \frac{\mu}{2(\mu n + L)}\right)^k\left[\left\|x^0 - x^*\right\|^2 + \frac{n}{\mu n + L}\left[f(x^0) - \left\langle f'(x^*), x^0 - x^*\right\rangle - f(x^*)\right]\right].$$

*Here the expectation is over all choices of index $j^k$ up to step $k$.*

## 6 Experiments

We performed a series of experiments to validate the effectiveness of SAGA. We tested a binary classifier on MNIST, COVTYPE, IJCNN1 and a least squares predictor on MILLIONSONG. Details of these datasets can be found in [9]. We used the same code base for each method, just changing the main update rule. SVRG was tested with the recalibration pass used every $n$ iterations, as suggested in [8]. Each method had its step size parameter chosen so as to give the fastest convergence.

We tested with a L2 regulariser, which all methods support, and with a L1 regulariser on a subset of the methods. The results are shown in Figure 2. We can see that Finito (perm) performs the best on a per epoch equivalent basis, but it can be the most expensive method per step. SVRG is similarly fast on a per epoch basis, but when considering the number of gradient evaluations per epoch is double that of the other methods for this problem, it is middle of the pack. SAGA can be seen to perform similar to the non-permuted Finito case, and to SDCA. Note that SAG is slower than the other methods at the beginning. To get the optimal results for SAG, an adaptive step size rule needs to be used rather than the constant step size we used. In general, these tests confirm that the choice of methods should be done based on their properties as discussed in Section 3, rather than their convergence rate.

## Footnotes

*The first author completed this work while under funding from NICTA. This work was partially supported by the MSR-Inria Joint Centre and a grant by the European Research Council (SIERRA project 239993).

[1]More precisely, to Option I of Prox-SDCA as described in [11, Figure 1]. We will simply refer to this method as "SDCA" in this paper for brevity.

[2]This is because $f_i^*(\alpha_i) = \inf_{\tilde{\alpha}_i \text{ s.t. } \alpha_i = X_i\tilde{\alpha}_i} \psi_i^*(\tilde{\alpha}_i)$.

[3]Note that the first equality below is the only place in the proof where we use the fact that $x^*$ is an optimality point.

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
