[Supplementary Material]

# Appendix

## A   The SDCA/Finito Midpoint Algorithm

Using Lagrangian duality theory, SDCA can be shown at step $k$ as minimising the following lower bound:

$$A^k(x) = \frac{1}{n} f_j(x) + \frac{1}{n} \sum_{i \neq j}^{n} \left[ f_i(\phi_i^k) + \langle f_i'(\phi_i^k), x - \phi_i^k \rangle \right] + \frac{\mu}{2} \|x\|^2 .$$

Instead of directly including the regulariser in this bound, we can use the standard strong convexity lower bound for each $f_i$, by removing $\frac{\mu}{2}\|x\|^2$ and changing the expression in the summation to $f_i(\phi_i^k) + \langle f_i'(\phi_i^k), x - \phi_i^k \rangle + \frac{\mu}{2}\|x - \phi_i\|^2$. The transformation to having strong convexity within the $f_i$ functions yields the following simple modification to the algorithm: $\phi_j^{k+1} = \mathrm{prox}_{(\mu(n-1))^{-1}}^{f_j}(z)$, where:

$$z = \frac{1}{n-1} \sum_{i \neq j} \phi_i^k - \frac{1}{\mu(n-1)} \sum_{i \neq j} f_i'(\phi_i^k).$$

It can be shown that after this update:

$$x^{k+1} = \phi_j^{k+1} = \frac{1}{n} \sum_i \phi_i^{k+1} - \frac{1}{\mu n} \sum_i f_i'(\phi_i^{k+1}).$$

Now the similarity to Finito is apparent if this equation is compared Equation 8: $x^{k+1} = \frac{1}{n} \sum_i \phi_i^k - \gamma \sum_{i=1}^{n} f_i'(\phi_i^k)$. The only difference is that the vectors on the right hand side of the equation are at their values at step $k+1$ instead of $k$. Note that there is a circular dependency here, as $\phi_j^{k+1} := x^{k+1}$ but $\phi_j^{k+1}$ appears in the definition of $x^{k+1}$. Solving the proximal operator is the resolution of the circular dependency. This mid-point between Finito and SDCA is interesting in it's own right, as it appears experimentally to have similar robustness to permuted orderings as Finito, but it has no tunable parameters like SDCA.

When the proximal operator above is fast to compute, say on the same order as just evaluating $f_j$, then SDCA can be the best method among those discussed. It is a little slower than the other methods discussed here, but it has no tunable parameters at all. It is also the only choice when each $f_i$ is not differentiable. The major disadvantage of SDCA is that it can not handle non-strongly convex problems directly. Although like most methods, adding a small amount of quadratic regularisation can be used to recover a convergence rate. It is also not adapted to use proximal operators *for the regulariser* in the composite objective case. The requirement of computing the proximal operator of each loss $f_i$ initially appears to be a big disadvantage, however there are variants of SDCA that remove this requirement, but they introduce additional downsides.

## B   Lemmas

**Lemma A1.** *Let $f$ be $\mu$-strongly convex and have Lipschitz continuous gradients with constant $L$. Then we have for all $x$ and $y$:*

$$f(x) \geq f(y) + \langle f'(y), x - y \rangle + \frac{1}{2(L-\mu)} \|f'(x) - f'(y)\|^2$$

$$+ \frac{\mu L}{2(L-\mu)} \|y - x\|^2 + \frac{\mu}{(L-\mu)} \langle f'(x) - f'(y), y - x \rangle .$$

*Proof.* Define the function $g$ as $g(x) = f(x) - \frac{\mu}{2}\|x\|^2$. Then the gradient is $g'(x) = f'(x) - \mu x$. $g$ has a Lipschitz gradient with constant $L - \mu$. By convexity, we have [1, Thm. 2.1.5]:

$$g(x) \geq g(y) + \langle g'(y), x - y \rangle + \frac{1}{2(L-\mu)} \|g'(x) - g'(y)\|^2 .$$

Substituting in the definition of $g$ and $g'$, and simplifying the terms gives the result. $\qquad\square$

**Lemma 1.** *Let $f(x) = \frac{1}{n}\sum_{i=1}^{n} f_i(x)$. Suppose each $f_i$ is $\mu$-strongly convex and has Lipschitz continuous gradients with constant $L$. Then for all $x$ and $x^*$:*

$$\langle f'(x), x^* - x \rangle \leq \frac{L-\mu}{L}\left[f(x^*) - f(x)\right] - \frac{\mu}{2}\|x^* - x\|^2 - \frac{1}{2Ln}\sum_i \|f_i'(x^*) - f_i'(x)\|^2 - \frac{\mu}{L}\langle f'(x^*), x - x^* \rangle.$$

*Proof.* This is a straight-forward corollary of Lemma A1, using $y = x^*$, and averaging over the $f_i$ functions. $\qquad\square$

**Lemma 2.** *We have that for all $\phi_i$ and $x^*$:*

$$\frac{1}{n}\sum_i \|f_i'(\phi_i) - f_i'(x^*)\|^2 \leq 2L\left[\frac{1}{n}\sum_i f_i(\phi_i) - f(x^*) - \frac{1}{n}\sum_i \langle f_i'(x^*), \phi_i - x^* \rangle\right].$$

*Proof.* Apply the standard inequality $f(y) \geq f(x) + \langle f'(x), y - x \rangle + \frac{1}{2L}\|f'(x) - f'(y)\|^2$, with $y = \phi_i$ and $x = x^*$, for each $f_i$, and sum. $\qquad\square$

**Lemma 3.** *It holds that for any $\phi_i^k$, $x^*$, $x^k$ and $\beta > 0$, with $w^{k+1}$ as defined in Equation 1:*

$$\mathbb{E}\left\|w^{k+1} - x^k - \gamma f'(x^*)\right\|^2 \leq \gamma^2(1 + \beta^{-1})\mathbb{E}\left\|f_j'(\phi_j^k) - f_j'(x^*)\right\|^2 + \gamma^2(1 + \beta)\mathbb{E}\left\|f_j'(x^k) - f_j'(x^*)\right\|^2$$
$$- \gamma^2\beta\left\|f'(x^k) - f'(x^*)\right\|^2.$$

*Proof.* We follow a similar argument as occurs in the SVRG proof [2] for this term, but with a tighter argument. The tightening comes from using $\|x + y\|^2 \leq (1 + \beta^{-1})\|x\|^2 + (1 + \beta)\|y\|^2$ instead of the simpler $\beta = 1$ case they use. The other key trick is the use of the standard variance decomposition $\mathbb{E}[\|X - \mathbb{E}[X]\|^2] = \mathbb{E}[\|X\|^2] - \|\mathbb{E}[X]\|^2$ three times.

$$\mathbb{E}\left\|w^{k+1} - x^k + \gamma f'(x^*)\right\|^2$$
$$= \mathbb{E}\left\|\underbrace{-\frac{\gamma}{n}\sum_i f_i'(\phi_i^k) + \gamma f'(x^*) + \gamma\left[f_j'(\phi_j^k) - f_j'(x^k)\right]}_{:= \gamma X}\right\|^2$$
$$= \gamma^2\mathbb{E}\left\|\overbrace{\left[f_j'(\phi_j^k) - f_j'(x^*) - \frac{1}{n}\sum_i f_i'(\phi_i^k) + f'(x^*)\right] - \left[f_j'(x^k) - f_j'(x^*) - f'(x^k) + f'(x^*)\right]}^{X}\right\|^2 + \gamma^2\left\|\overbrace{f'(x^k) - f'(x^*)}^{\mathbb{E}[X]}\right\|^2$$
$$\leq \gamma^2(1 + \beta^{-1})\mathbb{E}\left\|f_j'(\phi_j^k) - f_j'(x^*) - \frac{1}{n}\sum_i f_i'(\phi_i^k) + f'(x^*)\right\|^2$$
$$+ \gamma^2(1 + \beta)\mathbb{E}\left\|f_j'(x^k) - f_j'(x^*) - f'(x^k) + f'(x^*)\right\|^2 + \gamma^2\left\|f'(x^k) - f'(x^*)\right\|^2$$
$$\text{(use variance decomposition twice more):}$$
$$\leq \gamma^2(1 + \beta^{-1})\mathbb{E}\left\|f_j'(\phi_j^k) - f_j'(x^*)\right\|^2 + \gamma^2(1 + \beta)\mathbb{E}\left\|f_j'(x^k) - f_j'(x^*)\right\|^2 - \gamma^2\beta\left\|f'(x^k) - f'(x^*)\right\|^2.$$

$\qquad\square$

# C  Non-strongly-convex Problems

**Theorem 2.** *When each $f_i$ is convex, using $\gamma = \frac{1}{3L}$, we have for $\bar{x}^k = \frac{1}{k}\sum_{t=1}^{k} x^t$ that:*

$$\mathbb{E}\left[F(\bar{x}^k)\right] - F(x^*) \leq \frac{4n}{k}\left[\frac{2L}{n}\|x^0 - x^*\|^2 + f(x^0) - \langle f'(x^*), x^0 - x^* \rangle - f(x^*)\right].$$

*Here the expectation is over all choices of index $j^k$ up to step $k$.*

*Proof.* A more detailed version of this proof is available in [3]. We proceed by using a similar argument as in Theorem 1, but we add an additional $\alpha\|x^k - x^*\|^2$ together with the existing $c\|x^k - x^*\|^2$ term in the Lyapunov function.

We will bound $\alpha \left\| x^k - x^* \right\|^2$ in a different manner to $c \left\| x^k - x^* \right\|^2$. Define $\Delta = -\frac{1}{\gamma} \left( w^{k+1} - x^k \right) - f'(x^k)$, the difference between our approximation to the gradient at $x^k$ and true gradient. Then instead of using the non-expansiveness property at the beginning, we use a result proved for prox-SVRG [4, 2nd eq. on p.12]:

$$\alpha \mathbb{E} \left\| x^{k+1} - x^* \right\|^2 \leq \alpha \left\| x^k - x^* \right\|^2 - 2\alpha\gamma \mathbb{E} \left[ F(x^{k+1}) - F(x^*) \right] + 2\alpha\gamma^2 \mathbb{E} \left\| \Delta \right\|^2 .$$

Although their quantity $\Delta$ is different, they only use the property that $\mathbb{E}[\Delta] = 0$ to prove the above equation. A full proof of this property for the SAGA algorithm that follows their argument appears in [3].

To bound the $\Delta$ term, a small modification of the argument in Lemma 3 can be used, giving:

$$\mathbb{E} \left\| \Delta \right\|^2 \leq \left( 1 + \beta^{-1} \right) \mathbb{E} \left\| f'_j(\phi_j^k) - f'_j(x^*) \right\|^2 + (1 + \beta) \mathbb{E} \left\| f'_j(x^k) - f'_j(x^*) \right\|^2 .$$

Applying this gives:

$$\begin{aligned}
\alpha \mathbb{E} \left\| x^{k+1} - x^* \right\|^2 \leq \ & \alpha \left\| x^k - x^* \right\|^2 - 2\alpha\gamma \mathbb{E} \left[ F(x^{k+1}) - F(x^*) \right] \\
& + 2(1 + \beta^{-1})\alpha\gamma^2 \mathbb{E} \left\| f'_j(\phi_j^k) - f'_j(x^*) \right\|^2 + 2 \left( 1 + \beta \right) \alpha\gamma^2 \mathbb{E} \left\| f'_j(x^k) - f'_j(x^*) \right\|^2 .
\end{aligned}$$

As in Theorem 1, we then apply Lemma 2 to bound $\mathbb{E} \left\| f'_j(\phi_j^k) - f'_j(x^*) \right\|^2$. Combining with the rest of the Lyapunov function as was derived in Theorem 1 gives (we basically add the $\alpha$ terms to inequality (10) with $\mu = 0$):

$$\begin{aligned}
& \mathbb{E}[T^{k+1}] - T^k \\
\leq \ & \left( \frac{1}{n} - 2c\gamma \right) \left[ f(x^k) - f(x^*) - \left\langle f'(x^*), x^k - x^* \right\rangle \right] - 2\alpha\gamma \mathbb{E} \left[ F(x^{k+1}) - F(x^*) \right] \\
& + \left( 4(1 + \beta^{-1})\alpha L\gamma^2 + 2(1 + \beta^{-1})cL\gamma^2 - \frac{1}{n} \right) \left[ \frac{1}{n} \sum_i f_i(\phi_i^k) - f(x^*) - \frac{1}{n} \sum_i \left\langle f'_i(x^*), \phi_i^k - x^* \right\rangle \right] \\
& + \left( (1 + \beta)c\gamma + 2(1 + \beta)\alpha\gamma - \frac{c}{L} \right) \gamma \mathbb{E} \left\| f'_j(x^k) - f'_j(x^*) \right\|^2 .
\end{aligned}$$

As before, the terms in square brackets are positive by convexity. Given that our choice of step size is $\gamma = \frac{1}{3L}$ (to match the adaptive to strong convexity step size), we can set the three round brackets to zero by using $\beta = 1$, $c = \frac{3L}{2n}$ and $\alpha = \frac{3L}{8n}$. We thus obtain:

$$\mathbb{E}[T^{k+1}] - T^k \leq -\frac{1}{4n} \mathbb{E} \left[ F(x^{k+1}) - F(x^*) \right] .$$

These expectations are conditional on information from step $k$. We now take the expectation with respect to all previous steps, yielding $\mathbb{E}[T^{k+1}] - \mathbb{E}[T^k] \leq -\frac{1}{4n} \mathbb{E} \left[ F(x^{k+1}) - F(x^*) \right]$, where all expectations are unconditional. Further negating and summing for $k$ from 0 to $k - 1$ results in telescoping of the $T$ terms, giving:

$$\frac{1}{4n} \mathbb{E} \left[ \sum_{t=1}^{k} \left[ F(x^t) - F(x^*) \right] \right] \leq T^0 - \mathbb{E}[T^k].$$

We can drop the $-\mathbb{E} \left[ T^k \right]$ term since $T^k$ is always positive. Then we apply convexity to pull the summation inside of $F$, and multiply through by $4n/k$, giving:

$$\mathbb{E} \left[ F(\frac{1}{k} \sum_{t=1}^{k} x^t) - F(x^*) \right] \leq \frac{1}{k} \mathbb{E} \left[ \sum_{t=1}^{k} \left[ F(x^t) - F(x^*) \right] \right] \leq \frac{4n}{k} T^0.$$

We get a $(c + \alpha) = \frac{15L}{8n} \leq \frac{2L}{n}$ term that we use in $T^0$ for simplicity. $\qquad \square$

# D   Example Code for Sparse Least Squares & Ridge Regression

The SAGA method is quite easy to implement for dense gradients, however the implementation for sparse gradient problems can be tricky. The main complication is the need for just-in-time updating of the elements of the iterate vector. This is needed to avoid having to do any full dense vector operations at each iteration. We provide below a simple implementation for the case of least-squares problems that illustrates how to correctly do this. The code is in the compiled Python (Cython) language.

```python
import random
import numpy as np
cimport numpy as np

cimport cython
from cython.view cimport array as cvarray

# Performs the lagged update of x by g.
cdef inline lagged_update(long k, double[:] x, double[:] g, unsigned long[:] lag,
                          long[:] yindices, int ylen, double[:] lag_scaling, double a):

    cdef unsigned int i
    cdef long ind
    cdef unsigned long lagged_amount = 0

    for i in range(ylen):
        ind = yindices[i]
        lagged_amount = k—lag[ind]
        lag[ind] = k
        x[ind] += lag_scaling[lagged_amount]*(a*g[ind])

# Performs x += a*y, where x is dense and y is sparse.
cdef inline add_weighted(double[:] x, double[:] ydata , long[:] yindices, int ylen, double a):
    cdef unsigned int i

    for i in range(ylen):
        x[yindices[i]] += a*ydata[i]

# Dot product of a dense vector with a sparse vector
cdef inline spdot(double[:] x, double[:] ydata , long[:] yindices, int ylen):
    cdef unsigned int i
    cdef double v = 0.0

    for i in range(ylen):
        v += ydata[i]*x[yindices[i]]

    return v

def saga_lstsq(A,  double[:] b, unsigned int maxiter, props):

    # temporaries
    cdef double[:] ydata
    cdef long[:] yindices
    cdef unsigned int i, j, epoch, lagged_amount
    cdef long indstart, indend, ylen, ind
    cdef double cnew, Aix, cchange, gscaling

    # Data points are stored in columns in CSC format.
    cdef double[:] data = A.data
    cdef long[:] indices = A.indices
    cdef long[:] indptr = A.indptr

    cdef unsigned int m = A.shape[0] # dimensions
    cdef unsigned int n = A.shape[1] # datapoints
```

```python
cdef double[:] xk = np.zeros(m)
cdef double[:] gk = np.zeros(m)

cdef double eta = props['eta'] # Inverse step size = 1/gamma
cdef double reg = props.get('reg', 0.0) # Default 0
cdef double betak = 1.0 # Scaling factor for xk.

# Tracks for each entry of x, what iteration it was last updated at.
cdef unsigned long[:] lag = np.zeros(m, dtype='I')

# Initialize gradients
cdef double gd = -1.0/n
for i in range(n):
    indstart = indptr[i]
    indend = indptr[i+1]
    ydata = data[indstart:indend]
    yindices = indices[indstart:indend]
    ylen = indend-indstart
    add_weighted(gk, ydata, yindices, ylen, gd*b[i])

# This is just a table of the sum the geometric series (1-reg/eta)
# It is used to correctly do the just-in-time updating when
# L2 regularisation is used.
cdef double[:] lag_scaling = np.zeros(n*maxiter+1)
lag_scaling[0] = 0.0
lag_scaling[1] = 1.0
cdef double geosum = 1.0
cdef double mult = 1.0 - reg/eta
for i in range(2,n*maxiter+1):
    geosum *= mult
    lag_scaling[i] = lag_scaling[i-1] + geosum

# For least-squares, we only need to store a single
# double for each data point, rather than a full gradient vector.
# The value stored is the A_i * betak * x product
cdef double[:] c = np.zeros(n)

cdef unsigned long k = 0 # Current iteration number

for epoch in range(maxiter):

    for j in range(n):
        if epoch == 0:
            i = j
        else:
            i = np.random.randint(0, n)

        # Selects the (sparse) column of the data matrix containing datapoint i.
        indstart = indptr[i]
        indend = indptr[i+1]
        ydata = data[indstart:indend]
        yindices = indices[indstart:indend]
        ylen = indend-indstart

        # Apply the missed updates to xk just-in-time
        lagged_update(k, xk, gk, lag, yindices, ylen, lag_scaling, -1.0/(eta*betak))
```

```python
            Aix = betak * spdot(xk, ydata, yindices, ylen)

            cnew = Aix
            cchange = cnew—c[i]
            c[i] = cnew
            betak *= 1.0 — reg/eta

            # Update xk with sparse step bit (with betak scaling)
            add_weighted(xk, ydata, yindices, ylen, —cchange/(eta*betak))

            k += 1

            # Perform the gradient—average part of the step
            lagged_update(k, xk, gk, lag, yindices, ylen, lag_scaling, —1.0/(eta*betak))

            # update the gradient average
            add_weighted(gk, ydata, yindices, ylen, cchange/n)

    # Perform the just in time updates for the whole xk vector, so that all entries are up—to—date.
    gscaling = —1.0/(eta*betak)
    for ind in range(m):
        lagged_amount = k—lag[ind]
        lag[ind] = k
        xk[ind] += lag_scaling[lagged_amount]*gscaling*gk[ind]
    return betak * np.asarray(xk)
```