[Reviews · NeurIPS 2014]

Submitted by Assigned_Reviewer_1

This important paper introduces an incremental gradient method called SAGA for minimizing f+h when f is an average of n functions f_i (like ERM) and h is a possibly non-smooth regularizer (for which we know a prox operator), that has provable linear convergence rates when f (or each f_i according to the case) is strongly convex and the same algorithm without a change has a 1/k rate when strong convexity is absent, hence allowing it to adapt to the level of strong convexity in the problem, automatically interpolating between the rates.

One of the important contributions of the paper is to show the similarities between a vast array of existing algorithms for this problem like SAG (which did not provably handle prox operators and was non-adaptive to strong convexity), and SDCA (developed over the last few years), and more recent algorithms from the past year like MISO and SVRG. While the relationship to SAG is rather straightforward (as hinted by the name SAGA), it was very interesting to see the connections to MISO and SVRG and even more so SDCA.

One of the most interesting features of the paper was its ability to verbally explain why this method works well and how it relates to other methods. I found it rather interesting that it uses a biased gradient unlike SAG which results in a reduced variance. It is also interesting how SVRG is very similar to SAGA but trades off computation for space by not storing and updating the \phi vectors frequently. The relation to MISO/Finito is also interesting and not direct. Lastly, viewing the SDCA algorithm in the primal might also be of independent interest and even then it is unclear that there is any relation between the algorithms, so writing it in the same form as MISO is very interesting.

The main contribution of the paper is its simple and clear proofs of convergence (as compared to SAG!) for both the strongly convex and the non-strongly convex case in the appendix, which will be of use in understanding these algorithms as well as designing new ones. Since it is adaptive to the presence or absence of strong convexity, it seems practically extremely useful when it is not known what the strong convexity constant is.

Because of the importance of these kinds of problems in ML (ERM with regularizers with or without strong convexity), and the comparison to existing methods, I think this is a very important paper to many people in the community (while optimization people might be interested in the proofs, it also seems very practically applicable for everyone else).

Possible typos - It seems to me like the "s" in many places (description of MISO and in proof of Lemma 1) should actually be "mu". In the appendix "manor" should be "manner".
Summary: This paper presents an algorithm for minimizing (adaptive to possibly non-strongly convex) finite sums with regularizers with simple proofs of convergence and comparisons to existing algorithms in the literature.

I think this is an important paper that will definitely interest both theoreticians and practitioners. I congratulate the authors on the good effort spent in relating it to existing algorithms and presenting the proofs in an understandable manner. I highly recommend it for acceptance.

Submitted by Assigned_Reviewer_38

This paper proposes a new incremental gradient algorithm based on the spirit of several relative methods proposed recently. This work makes some variances on the update rule of gradient.

To my best of knowledge, the core idea of incremental gradient algorithm is to reduce the variance of the proximal gradient to get better convergence rate. From this kind of view, this work is not so attractive as the convergence rate doesn’t improve significantly and the idea of this paper is not very novel. This also can be seen from the result of experiments. The result of the method in this paper gets similar convergence speed with the other methods and doesn’t always outperform the existing methods.

Besides, there maybe some small errors in the proof, e.g., the equation in line 324 may miss a scalar eta.

However, this is a completed work with relatively simple theory analysis and the author clarifies clearly the relationship of SAGA with the other related works.
Summary: This is a completed work with relatively simple theory analysis. However, the main contribution of this work is not very attractive.

Submitted by Assigned_Reviewer_47

SAGA is a new incremental gradient technique. Purely in terms of performance, its salient features are that on strongly convex problems, it is provably faster than SAG and SVRG, and almost as fast as SDCA. The main advantage over SDCA is that the algorithm works without modification in the absence of strong convexity. “Without modification” here means without needing to explicitly add a regularizer to make the problem strongly convex. SAGA does seem to need to explicitly know the Lipschitz and strong convexity constants, in order to obtain linear convergence. Indeed, as discussed below, another advantage of SAGA is that it need only tune one step-size parameter (what is labelled \eta in the paper). It would be interesting to know how this is computed in practice when the values are not explicitly known.

Section 3 provides not only an explanation of SAGA, but also a nice summary of a few other popular methods. This is valuable to the paper, since many variants of these methods have been in use for some time, and understanding them better in a unified and compared setting is quite useful. For example, the discussion on SDCA is quite interesting. By a transformation of the formulation, the authors are able to show a much more clear connection to other primal based methods (in particular, MISOmu).

SAGA is an improvement over these methods because of better analysis (e.g., SAG), ability to deal with prox operators, fewer parameters that must be carefully tuned (e.g., compared to SVRG), and ability to deal with absence of strong convexity automatically.

Given that the ability to handle non-strongly convex composite functions automatically is a key selling point, the main point of the paper would be strengthened if the authors could explain, or even just show more clearly in computation, that adding a regularizer with a small constant and then using, e.g., SDCA, does not result in a better solution technique.

The paper seems well written. I only found errors on line 121 (effect should be affect) and then in 298, it seems a \mu turned into an s.
Summary: The paper is well written and the more unified exposition of several incremental gradient methods is in itself useful. The paper would be strengthened with a more clear differentiation from other algorithms -- how hard is it to tune two parameters instead of one? how much better is this algorithm in the absence of strong convexity, compared to others that require a regularizer?
Author Feedback
Author rebuttal: For reviewer 47, we would like to note that knowledge of both the strong convexity and Lipschitz constants is only needed to get the best rate when $n$ is small. We note on lines 91-92 that using a step length of $3L$ gives almost as good a rate, and requires no knowledge of the strong convexity constant.

We also agree that some clarification of the practicality of adding small regularization constants to the other methods should be included. We will add an additional experiment illustrating the issue. We will also add a discussion on how adding small regularization to the other methods requires more tuning than our method.

Reviewer 38 points out a missing $\eta$ constant on one line. This typo is fixed on the following line of the proof, so doesn't effect the correctness of the proof as a whole. We would like to argue that our contribution is more significant than reviewer 38 states, as our method is the first fast incremental gradient method that is automatically adaptive to strong convexity, supports non-strongly convex functions, and allows for the use of non-differentiable regularizers. We believe our overview of existing IG methods is also a valuable contribution to the literature.